# SparCas: A Dimension-First Cascade for Efficient Long-Context LLM Inference

## Abstract

Large language models (LLMs) have demonstrated strong capability in handling long-context sequences, but inference efficiency is bottlenecked by the continuously growing KV cache. KV cache selection methods mitigate this by retaining only "important" tokens for attention, yet existing solutions face a fundamental dilemma: they either rely on coarse page-level heuristics (sacrificing precision) or expensive token-wise approximation or scanning (sacrificing speed). We introduce **Sparsity Cascade (SparCas)**, a novel dimension-first cascade that resolves this trade-off. SparCas is grounded in the empirical discovery that token importance ranking is remarkably stable to dimension pruning. Leveraging this, we instantiate a ***prune-in-prune*** cascade: (i) *intra-token sparsity* first **prunes the feature space** to identify critical **dimension indices** using a lightweight query-only proxy, and (ii) *cross-token sparsity* then **prunes the context length** by using this **tiny subset of dimensions** to efficiently filter for salient **token indices**. This approach effectively decouples the cost of ranking from the context length. Across extensive evaluations on PG-19, LongBench, and RULER, SparCas consistently matches or outperforms dense attention and prior baselines, achieving oracle-level accuracy with budgets as few as $1\%$ of tokens at a 32K-token context. Integrated into FlashInfer, SparCas delivers up to $3.01\times$ faster self-attention and $1.64\times$ end-to-end speedups. Our project is anonymously available at https://anonymous.4open.science/r/sparcas/.

## 1 Introduction

Large language models (LLMs) such as LLaMA (Touvron et al., 2023) and GPT-4 (OpenAI, 2023) demonstrate exceptional efficacy in handling contexts of *hundreds of thousands* of tokens, significantly enhancing applications like richer dialogue systems and long-document reasoning tasks (Peng et al., 2023; Liu et al., 2024; Anthropic, 2024). However, the auto-regressive generation paradigm intrinsic to LLMs mandates the storage of key-value (KV) pairs computed during previous decoding steps to avoid re-computation (Vaswani et al., 2017; Pope et al., 2022). As the context window widens, the KV cache—not the model weights—rapidly dominates resources; for a 7B-parameter LLaMA at a 32K window, the cache alone occupies roughly 16 GB (FP16) and adds *tens of milliseconds* of HBM traffic per generated token, putting memory bandwidth—not compute—firmly on the critical path (Saxena et al., 2024). Therefore, it emerges as a valuable and crucial research topic for optimizing the memory efficiency of the KV cache while preserving decoding quality.

An intuitive solution to mitigate this inference bottleneck is *cache eviction*, *i.e.*, discarding the KV cache of less significant tokens once the cache reaches a predefined capacity budget. Existing sliding-window or FIFO policies (Beltagy et al., 2020; Xiao et al., 2023) prioritize retaining only the most recent tokens (and sometimes the initial "sink" tokens). More elaborate schemes track historical attention scores to disregard low-impact tokens (Zhang et al., 2023; Oren et al., 2024). While cache eviction appears intuitively effective, once discarded, a token can no longer be revisited. This limitation can lead to substantial degradation in inference quality if the relevance of that token resurfaces in subsequent computations (Tang et al., 2024).

In contrast, *cache selection* retains the full KV cache in memory but consults only a subset per step. Heuristics such as Quest's page-level scoring (Tang et al., 2024) or LSH bucketing (Kitaev et al., 2020) exploit the fact that attention is sparse—only a small fraction of historical KV cache matters

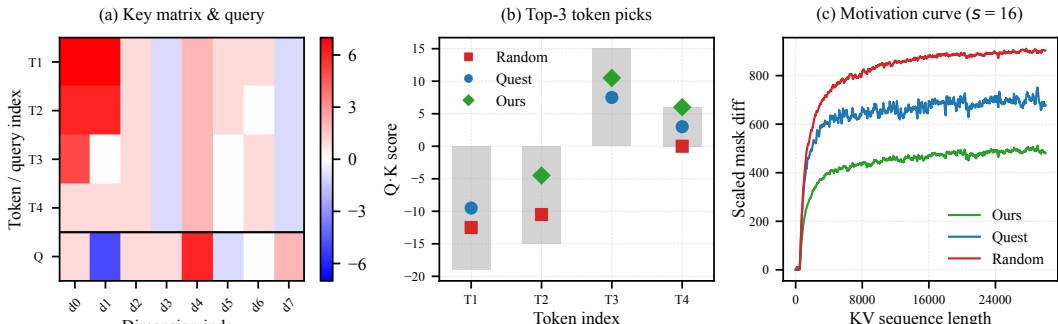

Figure 1: Hidden-dimension sparsity preserves token importance order. (a) Toy $4\times8$ key matrix $K$ and query $Q$. (b) Exact dot-product ($Q{\cdot}K$) scores for four tokens, ranked by ground truth (grey bars). Quest selects tokens using a coarse-grained page statistic and thus misses the correct top-3 tokens. In contrast, pruning six insignificant hidden dimensions does not affect the correct selection. (c) PG-19 validation, using hidden-dimension pruning to $s = 16$ per head. We measure token selection stability as the difference summarization in binary token-selection masks compared to the oracle mask selected by full attention score. Our approach consistently selects tokens most similar to the baseline, outperforming Quest and random selection.

for attention scores. However, these methods exhibit unique drawbacks for *on-the-fly* cache selection: Quest's page-level selection fails to capture the fine-grained variations in the importance of individual tokens, while token-level hashing introduces additional computational overhead. Effectiveness of cache selection therefore hinges on a notion of token *criticality* that is both *accurate* and *efficient*.

While oracle token ranking requires prohibitively expensive full attention computation, our key insight is that this ranking is remarkably stable to dimension pruning. As shown in Figure 1, the relative token order is preserved even when most key dimensions are discarded. This allows an approximate partial score, computed efficiently over a small dimensional subset, to serve as an accurate and practical heuristic for token selection (see Section 3.3).

Based on this motivation, we propose Sparsity Cascade (SPARCAS), a method that cascades two orthogonal forms of sparsity. First, it applies *intra-token sparsity* by pruning non-critical key dimensions using a lightweight, query-only heuristic. Second, it leverages the resulting compact key cache to perform *cross-token sparsity*, efficiently computing partial scores to select the most relevant tokens for attention. As illustrated in Figure 2, this cascade design yields a highly stable token ranking (Figure 1(c)) while drastically reducing the computational overhead compared to both page-level (Tang et al., 2024) and hashing-based (Chen et al., 2024) selection methods.

We experimentally validated the effectiveness of SPARCAS across a diverse spectrum of long-context tasks and LLM architectures. The results demonstrate that SPARCAS not only matches the accuracy of dense attention but also achieves significant reductions in the KV cache budget, surpassing prior selection methods. Moreover, when integrated with the FlashInfer library on an NVIDIA A100 80G GPU, SPARCAS delivers a $1.64\times$ end-to-end speedup over dense attention at a 32K context length, while maintaining baseline accuracy using up to $128\times$ smaller token budgets across various long-context benchmarks. Our key contributions are summarized as follows.

- We show empirically that many key dimensions have negligible influence on token ranking; pruning them preserves the top-K token order over a wide sparsity range.

- We propose SPARCAS, a cascaded KV cache manager that first prunes dimensions and then tokens, providing an efficient and accurate mechanism for identifying essential cache entries during inference.

- Extensive experiments across PG-19 (Rae et al., 2019), LongBench (Bai et al., 2023), and RULER (Hsieh et al., 2024) benchmarks, spanning models from 7B to 70B parameters, demonstrate the generality and effectiveness of our approach.

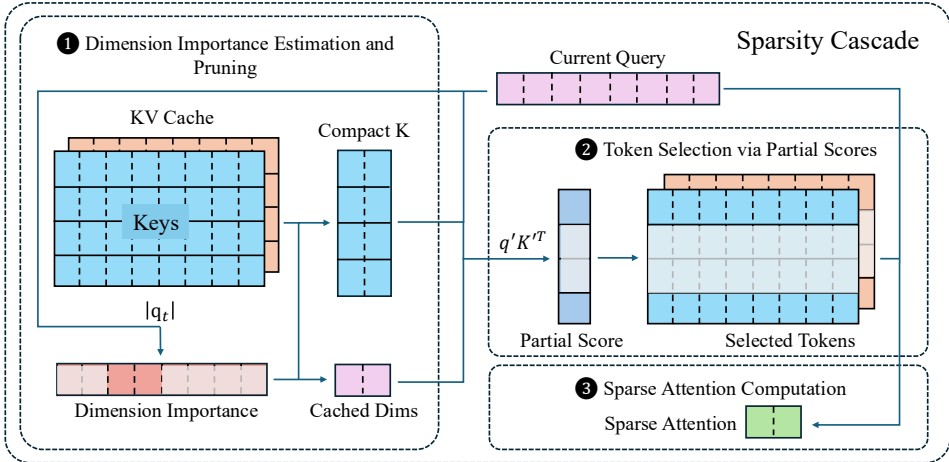

Figure 2: Workflow of SPARCAS. SPARCAS begins by estimating dimension importance and sparsifying the keys to their top-$s$ dimensions. It then computes partial attention scores using these sparse keys, followed by selecting the top-$K$ tokens most relevant to the current query for the final sparse attention computation.

## 2 RELATED WORK

### 2.1 LLM INFERENCE BOTTLENECKS

Transformer LLMs rely on a growing *key–value (KV) cache* to avoid recomputing past activations during autoregressive decoding (Pope et al., 2022). Its memory footprint can reach tens of GB for million-token prompts because the cache grows linearly with the length of the context (Peng et al., 2023; Liu et al., 2024).

However, large-scale inference faces distinct bottlenecks across the memory hierarchy. While *capacity* constraints can be mitigated by eviction policies (Zhang et al., 2023) or CPU-offloading (Lee et al., 2024), the latency of on-device decoding is strictly bound by **memory bandwidth**. Specifically, each decoding step must stream the cached tensors from High Bandwidth Memory (HBM) to compute units (Tang et al., 2024; Ye et al., 2024). Consequently, even there is enough capacity, the HBM transfer time dominates the critical path. Mitigating this specific on-device KV cache traffic is therefore the primary focus of this work.

### 2.2 KV CACHE EVICTION

One approach to managing the KV cache bottleneck is through *eviction* strategies, which reduce cache size by discarding entries deemed less important. Early methods often employed simple positional heuristics, such as sliding windows that retain only recent tokens (Beltagy et al., 2020) or techniques like StreamingLLM (Xiao et al., 2023) that preserve initial "sink" tokens alongside a recent window. More sophisticated strategies estimate token importance to guide this process: H2O (Zhang et al., 2023), for instance, retains historically high-scoring "heavy hitters," TOVA (Oren et al., 2024) uses the current query to decide which tokens to discard, and SnapKV (Sheng et al., 2024) periodically builds importance-based snapshots for eviction. While effective at bounding memory usage, all eviction methods suffer from the irreversible nature of deletion. Information discarded based on past interactions or simple heuristics might become critical later, potentially compromising performance on tasks requiring long-range dependencies or accurate recall (Tang et al., 2024). This risk makes eviction less suitable when preserving full contextual fidelity is essential.

## 2.3 KV CACHE SELECTION

A non-destructive alternative to eviction is *selection*, which retains the full cache but loads only a subset of entries for attention at each step. This paradigm aims to match dense attention accuracy while drastically reducing memory transfer. Success hinges on a selection proxy that is both accurate and efficient. While recent methods vary, their operational goal is universally to estimate the importance of each token or token-block directly.

The mechanisms for this estimation differ in three aspects: granularity, the proxy used to score importance, and the induced system cost. *Block/page-level* methods (for example, Quest (Tang et al., 2024)) precompute query-aware per-page statistics such as min/max over key dimensions and rank pages efficiently, but they can miss salient tokens inside a low-ranked page. *Token-level* methods operate on dense keys via hierarchical filters (PyramidKV (Cai et al., 2024), OmniKV (Hao et al., 2025)) or channel-pruned approximate attention scores (SparQ (Ribar et al., 2023)), and some use hashing to approximate nearest neighbors (MagicPIG (Chen et al., 2024)). These techniques aim to capture finer-grained signals; in practice, any gains come with added selection cost, since they still require per-token passes or maintain large host-side hash tables with CPU/GPU synchronization, so the overhead continues to scale with context length. *Learned proxies* (for example, NSA (Yuan et al., 2025)) train auxiliary heads to predict sparsity masks, which introduces architectural and training changes. *Precision-based* proxies (ZipCache (He et al., 2024)) quantize KV to a few bits and score on the quantized cache, reducing arithmetic and footprint but still scanning the cache each step. Across these methods, the proxy is typically computed by touching a representation of every token or block at each step, so bandwidth and latency grow roughly linearly with context length. A related but orthogonal systems line compresses or offloads the cache; for example, ShadowKV (Sun et al., 2025) keeps low-rank keys on the GPU, offloads values to the CPU, and reconstructs only needed pairs using landmarks and outliers. This primarily addresses capacity and placement rather than on-device bandwidth and complements our dimension-first SPARCAS.

SPARCAS replaces the standard *token-first* selection view with a novel *dimension-first* cascade. Instead of attempting to approximate the exact attention values for all tokens—which is computationally wasteful—SPARCAS leverages the empirical observation that the signal required to *rank* token importance is concentrated in a small subset of dimensions. Concretely, SPARCAS operates as a "prune-in-prun" pipeline. It begins with an *intra-token* pass, using a lightweight, query-only heuristic to identify high-magnitude channels. This filters out non-critical dimensions to form a compact "key sketch" dynamically tailored to the current step. Only then does it perform a *cross-token* pass, computing partial dot-products on this sketch. Crucially, because the system's goal is to identify the *set* of important tokens (ranking) rather than to recover their precise attention scores, these partial scores are sufficient to find the Top-K tokens with high fidelity. This dimension-first approach yields a superior efficiency–accuracy trade-off. By filtering dimensions *before* scanning tokens, it directly targets the memory-bandwidth bottleneck, converting the heavy-weight attention scan into a lightweight partial-score operation. Furthermore, SPARCAS preserves the original KV cache and requires no architectural changes, offering a seamless solution that efficiently isolates the most critical context.

## 3 METHODOLOGY

### 3.1 PRELIMINARY

Autoregressive inference in Transformer-based LLMs involves generating tokens sequentially. At each step $t$, the model computes attention using the current query vector $q_t$ alongside the Key ($K$) and Value ($V$) vectors corresponding to all preceding tokens $1, \ldots, t-1$. To avoid redundant computation, these past K and V vectors are stored in the KV cache. Standard dense attention requires loading the *entire* K and V cache from memory (typically HBM) to the compute units (e.g., GPU caches/registers) at every step, which becomes prohibitively expensive in terms of memory bandwidth for long-sequence generation (Pope et al., 2022; Tang et al., 2024).

KV cache *selection* methods aim to alleviate this bottleneck by exploiting the inherent sparsity often observed in attention mechanisms (Zhang et al., 2023; Ge et al., 2024). The core idea is to retain the full KV cache but, at each step $t$, dynamically select only a subset of important prior tokens with indices denoted as $I_t \subset \{1, \ldots, t-1\}$. Attention is then computed using only the KV cache

---

**Algorithm 1** SPARCAS: Cascaded Dimension & Token Selection

---

**Require:** Query $\boldsymbol{q}_t \in \mathbb{R}^{d_k}$, key cache $\{\boldsymbol{D}_b, \ldots, \boldsymbol{K}_{t-1}\}$, budgets $(D_b, T_b)$, update interval $U$
**Ensure:** Selected dimensions $\mathbb{D}_t$ ($|\mathbb{D}_t| = D_b$), tokens $\mathbb{I}_t$ ($|\mathbb{I}_t| = T_b$)
1: **if** $t \bmod U = 0$ **then**
2:   $\boldsymbol{s}_{\text{dim}} \leftarrow \text{score\_by\_q}(\boldsymbol{q}_t)$                          ▷ query-only per-channel score vector
3:   $\mathbb{D}_t \leftarrow \text{TOPINDICES}(\boldsymbol{s}_{\text{dim}}, D_b)$
4:   $\boldsymbol{K}_{\text{compact}} \leftarrow \boldsymbol{K}_{:,\,\mathbb{D}_t}$                          ▷ refresh compact key matrix (select columns)
5: **end if**
6: $\boldsymbol{q}'_t \leftarrow \boldsymbol{q}_{t,\,\mathbb{D}_t}$
7: $\boldsymbol{s}_{\text{tok}} \leftarrow \boldsymbol{K}^{\top}_{\text{compact}} \boldsymbol{q}'_t$                          ▷ partial scores over $D_b$ dims
8: $\mathbb{I}_t \leftarrow \text{TOPINDICES}(\boldsymbol{s}_{\text{tok}}, T_b)$
   **return** $(\mathbb{D}_t, \mathbb{I}_t)$

---

corresponding to this selected subset:

$$\text{Attention}(\boldsymbol{q}_t, \boldsymbol{K}, \boldsymbol{V}, \mathbb{I}_t) = \text{softmax}\left(\frac{\boldsymbol{q}_t \boldsymbol{K}^T_{\mathbb{I}_t,:}}{\sqrt{d_k}}\right) \boldsymbol{V}_{\mathbb{I}_t,:}$$

where $\boldsymbol{K}_{\mathbb{I}_t,:}$ and $\boldsymbol{V}_{\mathbb{I}_t,:}$ represent the rows of the Key and Value matrices corresponding to the indices in the set $\mathbb{I}_t$, and $d_k$ is the dimension of the key vectors. By choosing $|\mathbb{I}_t| \ll t-1$, the memory bandwidth required for loading $\boldsymbol{K}$ and $\boldsymbol{V}$ vectors and the attention computation cost can be significantly reduced. Therefore, the central challenge of KV cache selection lies in efficiently and accurately identifying the optimal subset $\mathbb{I}_t$ at each step.

### 3.2 CHALLENGES IN SELECTION HEURISTICS

Existing KV cache selection methods employ various heuristics to approximate the importance or *criticality* of past tokens (or groups of tokens) relative to the current query $\boldsymbol{q}_t$. For instance, Quest estimates page relevance using precomputed min/max key statistics, while LSH-based methods use hashing to group potentially relevant keys and queries.

Despite the advancement, these heuristics face an inherent trade-off between accuracy and computational overhead. Coarse-grained approximations like Quest's page-level statistics overlook important individual tokens within discarded pages. That is, a page considered important can contain numerous redundant tokens, whereas a page deemed unimportant may exhibit a few important tokens. As illustrated in Figure 1(c), the tokens selected by Quest and those identified by the oracle attention exhibit substantial discrepancies. Conversely, fine-grained selection methods like hashing the KV cache for token retrieval can introduce significant latency costs during the selection process itself, potentially eroding the speedup gained from sparse attention.

This challenge motivates a re-evaluation of the selection goal. While the optimal evaluation requires the computationally expensive oracle full attention score, we posit that a **simpler, relaxed objective** exists: only the *relative order* of the full attention scores is required, rather than precisely calculating these scores. Building on our key insight, we show that this token importance ranking is remarkably stable to dimension pruning, allowing an efficient approximation to effectively guide the selection process. As shown in Figure 1 (c), while the approximated scores may deviate slightly from the full attention scores, the Top-K ranked tokens remain largely consistent. This robustness suggests that an efficient approximation could effectively guide the token selection process without compromising attention precision.

### 3.3 THEORETICAL MOTIVATION: QUERY-DRIVEN INTERACTION SPARSITY

The robustness of token ranking to dimension pruning is grounded in the concentration of attention scores in high-dimensional spaces.

Consider the simplified attention score computation between a query vector $\boldsymbol{q}_t$ and a key vector $\boldsymbol{k}_j$ from the key matrix $\boldsymbol{K}$:

$$s_{t,j} = \boldsymbol{q}_t \cdot \boldsymbol{k}_j^\top = \sum_{d=1}^{d_k} \mathrm{q}_{t,d} \cdot \mathrm{k}_{j,d} \tag{1}$$

**Interaction Sparsity.** Our empirical analysis reveals that the query-key interaction terms $|q_{t,d} \cdot k_{j,d}|$ exhibit extreme statistical sparsity. The distribution is heavy-tailed with an average kurtosis of $\kappa \approx 68$, indicating that the final attention score is dominated by a very small subset of dimensions.

**Softmax Specialization.** We attribute this interaction sparsity to the intrinsic distribution of the query vectors. Transformer *activations* develop systematic "outlier features" with extreme magnitudes instead of following a standard Gaussian distribution, as established in quantization literature (Dettmers et al., 2022; Sun et al., 2024). We note that the projected query vectors $\boldsymbol{q}_t$ likewise show a heavy-tailed distribution, which is consistent with these results. This sparsity, we contend, is a learned adaptation to the Softmax mechanism. The model must produce large dot-product differences to saturate the Softmax function in order to successfully attend to a particular token among thousands of others. In order to accomplish this, the model effectively suppresses the non-essential dimensions to almost zero by concentrating massive energy into particular, distinguishing dimensions. Consequently, $|\boldsymbol{q}_t|$ acts as a naturally sparse, high-kurtosis selector.

**Theoretical Intuition.** This heavy-tailed distribution creates a multiplicative gating effect. We contend that the query vector $|\boldsymbol{q}_t|$ functions as a *dynamic feature selector*. It represents the model's current query intent. If the magnitude $|\boldsymbol{q}_{t,d}|$ for a particular dimension $d$ is low, acting as a dampener, rendering the corresponding key value $\boldsymbol{k}_{i,d}$ irrelevant to the final score, regardless of its magnitude. Conversely, high-magnitude query dimensions show where the model is concentrating its "energy." Because those dimensions are essentially "muted" by the query itself, pruning dimensions with low $|\boldsymbol{q}_{t,d}|$ is mathematically safe.

Formally, if we partition the dimensions into a critical set $\mathbb{D}_c$ and a non-critical set $\mathbb{D}_n$, the attention score can be decomposed as:

$$s_{t,j} = \underbrace{\sum_{d \in \mathbb{D}_c} \mathrm{q}_{t,d} \cdot \mathrm{k}_{j,d}}_{\text{dominant term}} + \underbrace{\sum_{d \in \mathbb{D}_n} \mathrm{q}_{t,d} \cdot \mathrm{k}_{j,d}}_{\text{negligible noise}} \tag{2}$$

When $|\mathbb{D}_c| \ll d_k$, the ranking of tokens based on the partial score computed over the dominant term closely approximates the ranking based on the full score, enabling accurate token selection with dramatically reduced computation.

### 3.4 SPARSITY CASCADE: A DIMENSION-FIRST APPROACH

SPARCAS addresses the KV cache bottleneck via a sequential, two-stage *prune-in-prune* process. It first applies *intra-token sparsity* to prune key dimensions based on the query, then uses this compact representation to perform *cross-token sparsity* to select the most relevant tokens. The overall workflow is formalized in Algorithm 1.

**Stage 1: Intra-Token Dimension Pruning.** Guided by the intuition in Section 3.3, we identify the critical dimension set $\mathbb{D}_c$ using the query magnitude. At each step, we rank key dimensions based solely on the current query vector $\boldsymbol{q}_t$:

$$S_{\mathrm{dim},d} = |\boldsymbol{q}_{t,d}|, \quad d \in \{1, \ldots, d_k\} \tag{3}$$

This zero-access heuristic is not only faster but more accurate than key-aware metrics, as it dynamically adapts to the head's current "interest" while filtering out static noise. Ablation studies (Table 2a) confirm its efficacy. We select the top-$D_b$ dimensions to form a set $\mathbb{D}_t$, which defines a compact key matrix $\boldsymbol{K}_{\mathrm{compact}} \in \mathbb{R}^{(t-1) \times D_b}$ containing only the most critical features. To amortize overhead, this dimension set is updated only periodically (every $U$ steps, validated in Table 2b).

**Stage 2: Cross-Token Selection and Final Attention.** Next, we use the persistent compact key matrix to efficiently compute partial attention scores for all past tokens:

$$\boldsymbol{s}_{\mathrm{tok}} = \boldsymbol{K}_{\mathrm{compact}} (\boldsymbol{q}_t')^\top \tag{4}$$

where $q_t'$ is the query vector projected onto the selected dimensions $\mathbb{D}_t$. Based on these scores, we select the top-$T_b$ tokens, forming a final index set $\mathbb{I}_t$. The attention output is then computed using the *full-dimensional* key and value vectors, $K_{\mathbb{I}_t,:}$ and $V_{\mathbb{I}_t,:}$, corresponding only to this small, dynamically identified subset of tokens.

**Memory and Bandwidth Efficiency.** This cascaded design is highly efficient. Stage 1 is a zero-cache-access operation. Stage 2 streams only the small $K_{\text{compact}}$ matrix, which is comparable in size to the metadata stored by page-level methods like Quest but enables true token-level resolution. The final attention step's memory traffic is capped by the budget $T_b$, reducing bandwidth by up to $\frac{L}{T_b}$ compared to dense attention. This entire pipeline operates on-device, eliminating the hashing, snapshotting, and CPU-GPU traffic required by other selection methods.

# 4 EXPERIMENTS

## 4.1 SETTING

**Benchmarks and Models.** To jointly cover long-prefill and long-decoding regimes, we evaluate on three complementary benchmarks. (1) **PG-19** (Rae et al., 2019) is a long-form language modeling corpus of public-domain books, with sequences exceeding 30K tokens. We use it as a *true long-decoding* benchmark and report cumulative perplexity over the full continuation, directly reflecting next-token prediction quality under long, continuous generation. (2) **LongBench** (Bai et al., 2023) is a heterogeneous suite of long-context tasks (open-domain QA, multi-document reading comprehension, code and math problems, summarization, and classification in multiple languages). Following prior work, we treat LongBench as an *approximate proxy* for long-prefill workloads: the model consumes a very long context but produces a short answer, and performance is measured with automatic metrics such as EM/F1/ROUGE with respect to gold answers. (3) **RULER** (Hsieh et al., 2024) consists of synthetic yet challenging retrieval and reasoning tasks with precisely controlled context length, evidence position, and distractors. Similar to LongBench, it approximates long-prefill behavior by stressing the model's ability to *locate and use* relevant information in a large cache while emitting a short response, which makes it well suited for studying memory-bandwidth–limited attention mechanisms.

Our evaluation spans multiple model families and scales, including LongChat-7B-v1.5-32k (Li et al., 2023), LLaMA-3.1-8B-Instruct (Meta AI, 2024), LLaMA-3.1-70B-Instruct, and Mistral-7B (Wu & Song, 2025), allowing us to test SparCas across both general-purpose and long-context–optimized LLMs.

**Baselines.** Our primary state-of-the-art baseline for accuracy comparison is Quest. For efficiency and latency measurements, we compare against a highly optimized Dense Attention implementation from FlashInfer (Ye et al., 2024). We also include accuracy results for MagicPIG on LongBench for additional context. However, we do not perform direct latency benchmarks against MagicPIG due to fundamental system-level differences; MagicPIG relies on a distinct architecture involving CPU-based LSH hashing and potential cache offloading, targeting a different efficiency trade-off (sampling with off-chip transfer) than our fully on-device, prune-in-prune cascade.

**SPARCAS Configuration.** Unless otherwise specified, SPARCAS is configured with a default dimension budget $D_b = 16$ (suggested by Table 3, Table 4, and Figure 6(b)), a token budget of $T_b = 2048$, and a dimension set update interval of $U = 64$ (suggested by Table 2b) steps. Following prior work, no selection or eviction mechanism is applied to the first two layers of the models. Early layers (0-2) are responsible for local feature extraction and possess high information density (lower kurtosis), making them less suitable for pruning. For Grouped Query Attention (GQA), dimension selection is performed per-group, maintaining the shared KV structure. Finally, we conduct efficiency benchmarks with a batch size of 1. This configuration is selected to isolate the memory bandwidth bottleneck, which is the primary constraint in real-time, on-device inference, and to provide a conservative evaluation by exposing the maximum relative impact of kernel launch overheads.

**Implementation.** Our implementation builds upon the FlashInfer and Quest libraries, incorporating optimized CUDA kernels for all stages of the SPARCAS pipeline. All experiments were conducted on NVIDIA A100 80G GPUs.

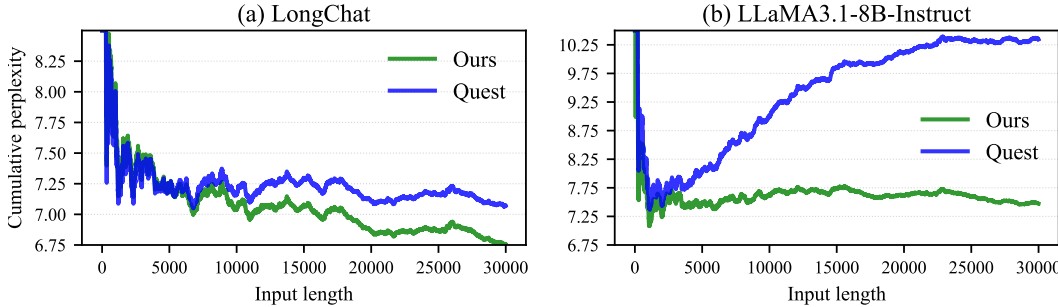

Figure 3: PG-19 Language modeling on LongChat-7B-v1.5-32k and LLaMA-3.1-8B-Instruct.

Table 1: Accuracy on the RULER benchmark. SPARCAS shows superior performance, scalability, and generalizability, especially in low-budget regimes.

| Model | Method | 128 | 256 | 512 | 1K | 2K | 4K | Full Cache |
|---|---|---|---|---|---|---|---|---|
| LLaMA3.1-8B-Instruct | QUEST | – | – | – | 85.15 | 90.29 | 92.53 | 94.14 |
| | SPARCAS | – | – | – | **86.99** | **92.20** | **93.28** | |
| LLaMA3.1-70B-Instruct | QUEST | 81.12 | 86.91 | 92.08 | 94.78 | 95.51 | 96.24 | 96.47 |
| | SPARCAS | **94.04** | **95.47** | **95.74** | **96.29** | **96.58** | **96.67** | |
| Mistral-7B | QUEST | – | – | – | 74.73 | 81.33 | 86.57 | 92.06 |
| | SPARCAS | – | – | – | **90.09** | **92.32** | **92.36** | |

## 4.2 ACCURACY EVALUATION

**Language Modeling on PG-19.** We first evaluate perplexity on the PG-19 test set, which comprises long documents suitable for assessing extended context performance. Using LongChat-7B-v1.5-32k and LLaMA-3.1-8B-Instruct, we process sequences up to 32K tokens and measure cumulative perplexity (lower is better) against Quest. As shown in Figure 3, SPARCAS consistently achieves lower perplexity than Quest across the full context length, suggesting its cascaded selection mechanism more effectively preserves critical information for next-token prediction.

**Performance on LongBench.** We validate performance on six diverse LongBench tasks using LLaMA-3.1-8B-Instruct. As detailed in Figure 4, SPARCAS consistently outperforms Quest across all tested token budgets. Notably, SPARCAS achieves near-lossless performance comparable to the full-cache baseline with significantly smaller token budgets (e.g., 1024-2048 tokens) than Quest, highlighting the superior accuracy-efficiency trade-off of our dimension-first approach.

**Scalability and Generalizability on RULER.** To evaluate scalability and generalizability, we extended experiments to the RULER benchmark with larger and more diverse models: LLaMA-3.1-70B-Instruct and Mistral-7B. As shown in Table 1, SPARCAS consistently outperforms Quest across models, with the most pronounced gains in low-budget regimes where efficient selection is critical. On the 70B model, SPARCAS remains highly effective even under extreme compression: at just 128 tokens (0.2% of the cache) it already achieves 94.04, substantially outperforming Quest (81.12). With only 256 tokens, SPARCAS surpasses 95, already within 99% of full-cache oracle accuracy, while Quest requires 2K tokens to approach the same level. This demonstrates SPARCAS's sharper ability to isolate salient content at far smaller budgets. Furthermore, strong performance on Mistral-7B confirms that the benefits of our dimension-first cascade generalize beyond the LLaMA family to different model architectures. Together, these results validate that SPARCAS is not only scalable to large models but also robust across architectures—providing a practical solution for efficient long-context inference.

## 4.3 EFFICIENCY EVALUATION

As long-context inference is memory-bound, SPARCAS's efficiency gain comes from reducing data movement by loading only a subset of the KV cache during attention computation. We quantify the performance of SPARCAS through kernel-level analysis and end-to-end latency measurements, eval-

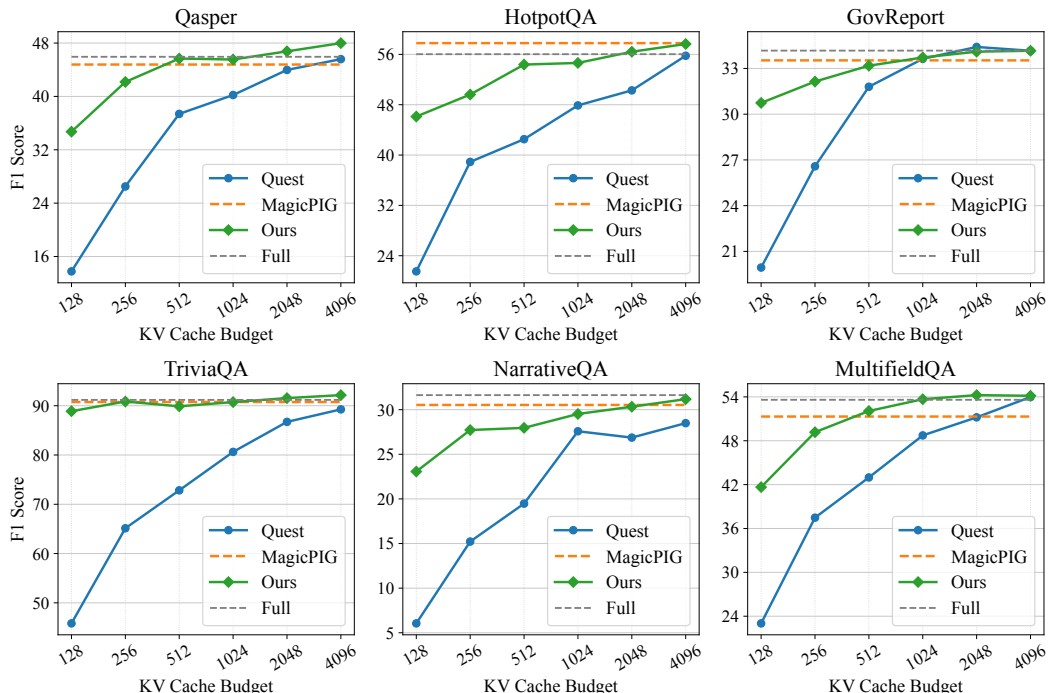

Figure 4: Performance evaluation on six LongBench datasets using the LLaMA-3.1-8B-Instruct model.

uated on NVIDIA A100 80G GPUs using NVBench (NVIDIA, 2024). Throughput scales inversely with per-token latency at fixed batch.

**Kernel-level breakdown.** Figure 5 contrasts the linear complexity of dense FlashInfer against the sublinear scaling of SPARCAS. The dense baseline is dominated by a single monolithic GEMM whose latency scales linearly $O(L)$ with context length. SPARCAS decomposes this monolith into lightweight kernels that effectively decouple computational cost from the full sequence length $L$:

**Top-Dim filtering** — executed every $U = 64$ steps, this step inspects *only the query*, ranks the channels by $|q_t|$, and caches the best $D_b \ll d_k$; its amortized cost is negligible. **Compact-K gathering** — a one-off pass that run immediately after prefill copies the selected $D_b$ channels for all existing tokens into a side cache, eliminating further full-cache reads. **Partial-score GEMM** — at every decode step, although scanning all $L$ past tokens to compute ranking scores, it operates on the tiny projected subspace $D_b \ll d_k$; only a $(t-1) \times D_b$ dot product is computed; because $D_b \ll d_k$, this kernel is $d_k/D_b$ times more efficient than the dense GEMM. With $D_b = 16$ and $d_k = 128$, this reduces memory traffic by $8\times$ compared to a full key scan. **Token filtering & sparse attention** — a tiny 1-D TOP-K picks the $T_b$ most relevant tokens, after which full-precision KV are loaded *only* for those tokens, capping memory traffic at $2T_bM$ bytes instead of $2LM$ bytes.

As a result, SPARCAS transforms the latency curve from linear to **sublinear**. As shown in Figure 6(a), the speedup factor improves as the context grows, reaching 1.79x at 8K, 2.47x at 16K, and 3.01x at 32K. This confirms that our overhead is constant while our savings scale with context length.

**End-to-end latency.** We benchmark both *prefill* and *decode* time on LongChat-7B-32k (FP16, single A100-80GB), taking FlashInfer as the full cache baseline. Although prefill must embed the whole prompt and therefore grows with sequence length for *all* methods, SPARCAS adds $< 4\%$ overhead because Stage 1 executes only once per prompt during prefill. Figure 6(a) plots the more critical *decode* latency: FlashInfer rises from $13.6$ ms at 4K to $35.1$ ms at 32K, re-streaming the full KV cache each step.

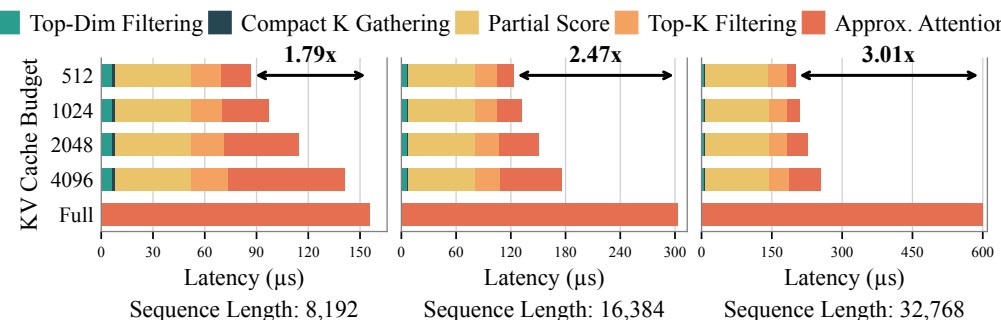

Figure 5: SPARCAS self-attention time breakdown compared to a dense FlashInfer baseline with $D_b = 16$. Stacked bars show the latency contribution of each SPARCAS stage: Top-Dimension Filtering (periodic, amortized), Compact K Gathering (one-time post-prefill), Partial Score computation, Top-K Token Filtering, and the final Approximate (Sparse) Attention.

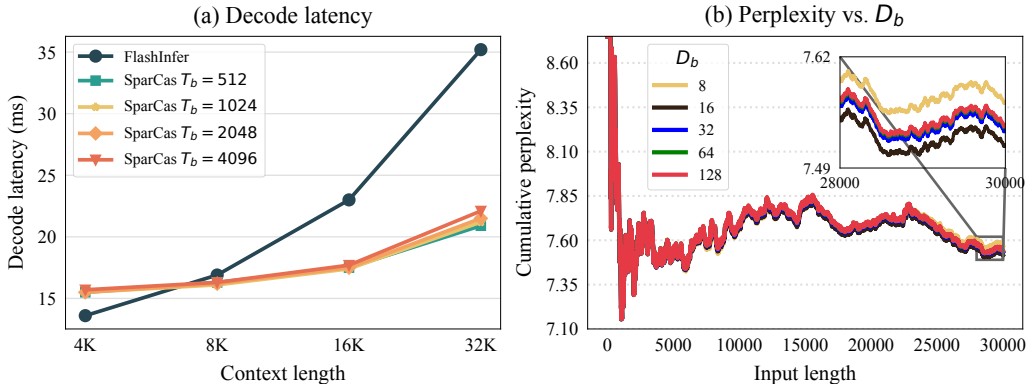

Figure 6: (a) Decode latency. SPARCAS stays flat after the context exceeds its fixed token budget $T_b$, while dense FlashInfer grows almost linearly. (b) Perplexity vs. dimension budget $D_b$. A tiny budget of 16 channels already matches the 128-dimensional baseline; even $D_b=8$ incurs only a small loss.

SPARCAS, in contrast, flattens once the context exceeds its token budget. With $(D_b, T_b) = (16, 512)$, it is $1.64\times$ faster at 32K. Even with token budgets up to 4K, latency stays within 5% of that baseline—demonstrating that throughput is *robust to budget choice*.

## 5 CONCLUSION

This work introduced SPARCAS, a novel selection method that addresses the KV cache bottleneck in long-context LLMs through a *dimension-first* cascaded sparsity paradigm. We grounded this approach in the discovery of a profound structural truth about attention mechanisms that *token importance ranking* is remarkably stable to dimension pruning. By first pruning non-critical feature dimensions and then selecting tokens from the resulting compact representation, SPARCAS efficiently identifies salient context with minimal overhead. Our extensive evaluations demonstrate that this approach achieves state-of-the-art performance, matching the accuracy of dense attention with a fraction of the token budget and delivering significant end-to-end speedups.

SPARCAS opens promising avenues for future research. First, dynamically adapting dimension and token budgets $(D_b, T_b)$ per layer or based on query complexity could further optimize resource usage. Furthermore, integrating sparsity awareness into LLM training or fine-tuning could enhance performance by promoting concentrated information distribution. Finally, the structured cascaded sparsity of SPARCAS lends itself to hardware co-design, enabling potential acceleration gains. Continued exploration of such hardware-friendly cache selection mechanisms will be critical as LLMs scale to handle ever-expanding contextual horizons.

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

# A APPENDIX

## A.1 ABLATION: SENSITIVITY TO DESIGN CHOICES

Here, we perform additional ablation studies of SPARCAS, including the dimension budget, dimension-selection metric, and the update interval for intra-token sparsity. All experiments in Table 2 are performed on the PG-19 validation split with the LLaMA-3-8B-Instruct ($PPL_1$) and longchat-7b-v.15-32k ($PPL_2$). Table 3 and Table 4 use RULER. We keep the token budget fixed at $T_b=2048$ (unless otherwise noted), set the maximum input length to 32K tokens, and apply SPAR-CAS from layer 3 onwards (layers 0–2 stay dense).

**Dimension budget.** Figure 6 (b) demonstrates that SPARCAS needs only $D_b=16$ channels to match the 128-dimensional baseline on LLaMA-3.1-8B-Instruct; $D_b=8$ raises perplexity by merely 0.05, and larger budgets give no extra gain—most attention signal lives in a very small sub-space.

Table 2: Ablation on dimension-selection metric and update interval $U$.

(a) Perplexity vs. dimension selector

| Metric | $PPL_1 \downarrow$ |
|---|---|
| $|\boldsymbol{q}_t| + \text{mean}(|\boldsymbol{K}|)$ | 7.473 |
| $|\boldsymbol{q}_t|$ | 7.470 |
| $\text{mean}(|\boldsymbol{K}|)$ | 7.581 |
| $|\boldsymbol{q}_t| + |\boldsymbol{K}|$ (token-wise) | 7.480 |
| Random | 115.0 |

(b) Perplexity vs. update gap $U$

| Interval $U$ | $PPL_1 \downarrow$ | $PPL_2 \downarrow$ |
|---|---|---|
| 1 | 7.470 | 6.757 |
| 16 | 7.490 | 6.759 |
| 64 | 7.500 | 6.754 |
| 1 024 | 7.552 | 6.757 |
| 2 048 | 7.551 | 6.760 |
| 4 096 | 7.625 | 6.783 |

Table 3: LLaMA-3.1-70B-Instruct performance with varying dimension budget ($T_b=2048$).

| $D_b$ | cwe | fwe | niah_mk1 | niah_mq | qa_1 | AVG |
|---|---|---|---|---|---|---|
| 2 | 77.4 | 97.3 | 96.0 | 94.0 | 58.0 | 84.6 |
| 4 | 94.8 | 98.0 | 98.0 | 99.0 | 80.0 | 94.0 |
| 8 | 96.4 | 96.0 | 98.0 | 100.0 | 86.0 | 95.4 |
| 16 | 97.6 | 96.0 | 100.0 | 100.0 | 86.0 | 96.0 |

Table 4: Mistral-7B-512K performance with varying dimension budget ($T_b=2048$).

| $D_b$ | cwe | fwe | niah_mk | niah_mq | niah_ | niah_si | qa_1 | AVG |
|---|---|---|---|---|---|---|---|---|
| 2 | 27.6 | 88.7 | 8.0 | 0.5 | 1.0 | 98.0 | 48.0 | 38.8 |
| 4 | 39.0 | 94.7 | 48.0 | 20.5 | 21.5 | 100.0 | 64.0 | 55.4 |
| 8 | 59.8 | 96.7 | 96 | 91.5 | 82.5 | 100.0 | 84.0 | 87.2 |
| 16 | 76.6 | 94.0 | 100.0 | 99.0 | 92.5 | 100.0 | 82.0 | 92.0 |

**Dimension-selection metric.** Table 2a shows that the *query-only* metric $|\boldsymbol{q}_t|$ achieves the best perplexity (7.470) while avoiding any K-cache reads. Adding the mean-key term brings no benefit and costs an extra pass; a token-wise variant is eight times slower yet equally accurate, and keys alone hurt quality. We therefore adopt $|\boldsymbol{q}_t|$ as our default scorer (see Section 3.3 for details).

**Update interval $U$.** Refreshing the dimension set every step ($U{=}1$) is wasteful: $U{=}64$ preserves accuracy and cuts Stage 1 overhead by $64\times$. Results stay stable up to $U{=}1024$; smaller gaps show diminishing returns; very large gaps ($U{=}4096$) begin to drift (Table 2b), so we set $U{=}64$ throughout, which is further validated by the performance observed on LongBench and RULER.

**Cross-model stability.** Tables 3 and 4 demonstrate that dimension importance concentrates similarly across models: budgets of $D_b{=}8$–16 suffice to match or exceed dense baselines in both 70B-scale and 7B-scale settings, supporting the generality of the dimension-first cascade.

## A.2 DISCUSSION: CONSTRUCTION ERROR AND MASK DIFFERENCES

This section presents an ablation study analyzing the impact of the dimension budget ($D_b$) on the fidelity of the approximate attention mechanism. We focus on two key metrics: *Construction Error* and *Mask Difference*.

### A.2.1 DEFINITIONS

**Construction Error (Relative Reconstruction Error).** Construction Error quantifies the accuracy of the approximate attention scores computed using the pruned dimensions compared to the exact scores computed using all dimensions. It is defined as the mean absolute error (MAE) between the full attention scores $\boldsymbol{A}$ and the partial attention scores $A_{D_b}$:

$$\text{Construction Error} = \frac{1}{N}\sum_{i=1}^{N}\big|\boldsymbol{A}_i - \boldsymbol{A}_{D_b,i}\big|, \tag{5}$$

where $\boldsymbol{A} = \frac{\boldsymbol{Q}\boldsymbol{K}^\top}{\sqrt{D}}$ is the standard attention score matrix, and $\boldsymbol{A}_{D_b} = \frac{\boldsymbol{Q}_{\text{sel}}\boldsymbol{K}_{\text{sel}}^\top}{\sqrt{D}}$ is the score matrix computed using only the selected top-$D_b$ dimensions. A lower error indicates that the reduced-dimension dot product effectively preserves the magnitude relationships of the attention scores.

**Mask Difference (Mask Diff).** Mask Difference measures the disagreement between the set of tokens selected by the approximate method and the set of tokens selected by the "oracle" method (using full dimensions). For a fixed token budget $T_b$, it is defined as the fraction of positions in the attention matrix where the binary masks differ:

$$\text{Mask Diff} = \frac{1}{N}\sum_{i=1}^{N}\mathbb{1}\!\!\!\;/\big(\boldsymbol{M}_{T_b,\text{oracle},i} \neq \boldsymbol{M}_{T_b,\text{approx},i}\big), \tag{6}$$

where $\boldsymbol{M}_{T_b,\text{oracle}}$ selects the top-$T_b$ tokens based on exact scores $\boldsymbol{A}$, and $\boldsymbol{M}_{T_b,\text{approx}}$ selects the top-$T_b$ tokens based on partial scores $\boldsymbol{A}_{D_b}$. This metric serves as a proxy for the accuracy of the heavy-hitter identification process.

### A.2.2 EXPERIMENTAL SETUP

We collect metrics using the `LLaMA-3.1-8B-Instruct` model evaluated on the PG-19 test set. The evaluation setup is:

- **Token budget** $T_b$: fixed at 512.
- **Dimension budget** $D_b$: varied across $\{2, 4, 8, 16, 32, 64, 128\}$.
- **Head dimension** $D$: 128 (full dimension for Llama-3).
- **Layers**: metrics are recorded for all attention layers (Layers 2–31, as Layers 0–1 use full-attention fallback).

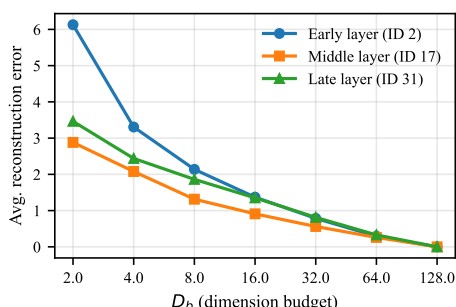

Figure 7: Average Reconstruction Error vs. dimension budget $D_b$ for selected layers. Early layers (e.g., Layer 2) show significantly higher error than middle or late layers at low $D_b$, indicating higher dimensionality in their attention features.

### A.2.3  CONSTRUCTION ERROR VS. DIMENSION BUDGETS

Figure 7 reports the construction error as a function of the dimension budget $D_b$. The error decreases monotonically as $D_b$ increases.

**Layer sensitivity.**  Early layers (e.g., Layer 2) exhibit significantly higher reconstruction error compared to middle layers, especially at low $D_b$. For instance, at $D_b = 2$, the error in Layer 2 (6.13) is more than double that of Layer 17 (2.88). This suggests that attention features in early layers are more distributed across dimensions, making them harder to compress.

**Convergence.**  As $D_b$ approaches $D = 128$, the error converges to zero, which is expected since $D_b = 128$ is mathematically equivalent to full attention.

### A.2.4  MASK DIFFERENCES VS. DIMENSION BUDGETS

Figure 8 shows the mask difference metric. The trend mirrors the construction error: better score approximation leads to more accurate token selection.

**High fidelity at low $D_b$.**  With a very small dimension budget of $D_b = 16$ (only $6.25\%$ of dimensions), the mask difference using $|q_t|$ proxy is smaller compared to the mask differences using other proxies or quest. This indicates that the heavy hitters in the attention matrix are largely determined by a small subset of highly informative dimensions; and the heavy hitters can be captured by the $|q_t|$ proxy.

**Robustness.**  The relatively low mask difference across layers suggests that the proposed dimension-pruned method is robust in identifying the most relevant tokens for attention, even when the exact score values are approximated.

### A.3  DISCUSSION: COMPATIBILITY WITH GROUPED-QUERY ATTENTION (GQA)

The SPARCAS heuristic is fully compatible with *Grouped-Query Attention* (GQA), which is now the default in many modern LLMs (for example, LLaMA 3.1). In GQA, multiple query heads share a single KV head (or a small number of KV heads). After projection, the KV representations are effectively replicated across a group of query heads, so that several queries attend to the same KV cache. From an inference and systems perspective, the resulting KV dataflow is very similar to standard multi-head attention (MHA): the main difference is that KV are shared across query heads instead of being strictly one-to-one.

SPARCAS is a non-destructive selection mechanism that operates on the KV cache without modifying its values. Under GQA, we simply apply the selection at the group level. For each GQA group, that is, the set of query heads that share the same KV cache, SPARCAS chooses a single shared subset of $D_b$ dimensions using our heuristic (for example, based on group-wise importance scores).

The same dimension mask is then reused for all query heads in that group. As a result, the KV cache continues to be stored once per GQA group, the selected $D_b$-dimensional subspace is defined once per group, and all query heads in the group attend using this common pruned subspace. This design mirrors other GQA-aware sparse attention methods and preserves the memory and compute advantages of GQA: we neither introduce additional KV copies nor break the sharing pattern.

From an engineering perspective, integrating SPARCAS into a GQA implementation is largely a matter of where the dimension selection is computed and stored. The cache layout remains the standard GQA layout with shared KV per group. During selection, we compute group-wise importance statistics and store a single dimension mask for each group. During lookup, every query head in the same group uses this mask to access the pruned KV cache. In practice, this keeps the kernel interface close to an existing GQA implementation and makes SPARCAS straightforward to integrate into GQA-based models such as LLaMA 3.1, while still delivering the intended memory-bandwidth benefits of selective cache access.

### A.4 COMPARISON WITH EVICTION-BASED BASELINES

In this section, we provide a direct comparison against state-of-the-art KV cache *eviction* methods, including PyramidKV (Cai et al., 2024), SnapKV (Sheng et al., 2024), and Ada-SnapKV (Feng et al., 2024). Unlike SPARCAS, which selects data dynamically from the full cache, these methods permanently discard tokens to save memory.

Table 5 reports the performance on the LongBench benchmark using the `LLaMA-3.1-8B-Instruct` model. All methods are restricted to a strict token budget of $T_b = 2048$ (excluding system prompts).

Table 5: LongBench performance comparison between SPARCAS and eviction-based baselines on LLaMA-3.1-8B-Instruct (Budget $T_b = 2048$). SPARCAS achieves the highest average accuracy. Bold indicates the best performance among compressed methods.

| Dataset | Full Cache | PyramidKV | SnapKV | Ada-SnapKV | SPARCAS |
|---|---|---|---|---|---|
| Qasper | 45.47 | 43.66 | 43.64 | 44.09 | **46.76** |
| HotpotQA | 55.97 | 55.06 | 54.24 | 54.80 | **56.42** |
| GovReport | 35.12 | 27.27 | 27.80 | 27.75 | **34.10** |
| TriviaQA | 91.64 | 91.35 | 91.65 | 91.64 | **91.95** |
| NarrativeQA | 30.22 | 29.81 | 29.75 | **30.64** | 30.33 |
| MultifiledQA | 55.80 | 54.88 | 55.06 | 55.46 | **56.24** |
| **Average** | 52.37 | 50.34 | 50.36 | 50.73 | **52.63** |

**State-of-the-Art Accuracy.** SPARCAS achieves the highest average accuracy (**52.63**) among all compressed methods, outperforming the Full Cache baseline (52.37). It beats Ada-SnapKV (50.73) and SnapKV (50.36) by effectively identifying salient tokens that heuristic-based eviction might discard.

**Robustness to Query Position.** A critical limitation of SnapKV-style methods (including PyramidKV and Ada-SnapKV) is their reliance on "recent window" heuristics. These methods assume that the most relevant information is either clustered in the heavy hitters or located in the most recent tokens. This makes them sensitive to the position of the query; if a query requires retrieving information from a "middle" section that was not flagged as a heavy hitter during prefill, the information is lost permanently. In contrast, SPARCAS employs a content-driven dimension proxy that does not depend on positional heuristics. This allows SPARCAS to dynamically recover relevant context regardless of its position in the sequence, resulting in superior robustness for tasks like *GovReport* (+3.23 over Ada-SnapKV) where evidence is scattered throughout the document.

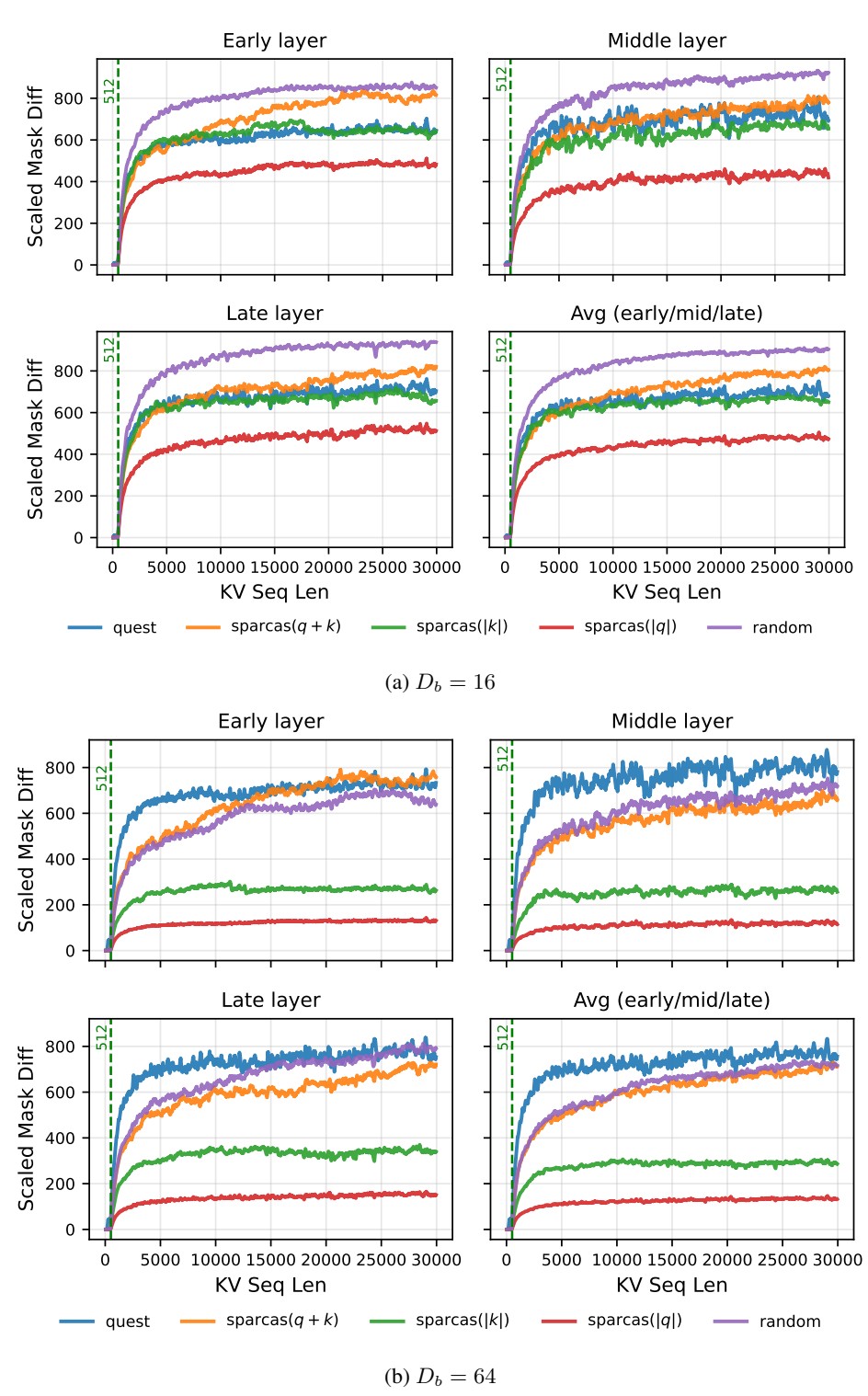

Figure 8: Scaled Mask Difference across sequence length for dimension budgets $D_b \in \{16, 64\}$. The green dashed line indicates the token budget ($T_b = 512$); higher values indicate greater divergence from the oracle selection.

