# OpenReview forum: "SparCas: A Dimension-First Cascade for Efficient Long-Context LLM Inference"
_ICLR.cc/2026/Conference — ICLR 2026 Conference Desk Rejected Submission_

### Official Review · Reviewer_fF7j · 2025-10-31

**Soundness:** 3
**Presentation:** 3
**Contribution:** 2
**Rating:** 4
**Confidence:** 4

**Summary:**

This paper introduces SPARCAS, a training-free, cascaded KV-cache selection method for long-context LLM inference. It exploits intra-token sparsity by pruning low-magnitude key dimensions via a lightweight query-only heuristic, then applies cross-token sparsity by computing partial scores in the reduced subspace to select top-K tokens. Integrated with FlashInfer, SPARCAS achieves near-dense accuracy on PG-19, LongBench, and RULER while delivering up to 1.6x+ end-to-end speedups on long sequence.

**Strengths:**

1. Simple but effective heuristic, the design focuses on reducing HBM bandwidth, the true bottleneck in long-context inference. The latency breakdown clearly isolates wins from partial-score GEMM and sparse attention.
2. Goo low-budget performance on RULER comparing to Quest, maintains high accuracy even at extreme compression, outperforming Quest dramatically.
3. Periodic dimension updates reduce overhead without noticeable quality degradation.

**Weaknesses:**

1. Layer/task dependence. The paper applies SPARCAS only from layer 3 but does not quantify how attention concentration, or optimal dimension set, varies across layers or tasks.
2. Several important families are absent from experiments, such as ShadowKV, SnapKV, H2O, while discussed conceptually, the empirical comparison is narrow, mostly vs Quest and MagicPIG.
3. Tensor-parallel integration. The paper does not explain how Dt is synchronized across TP shards, this an important omission for multi-GPU inference.

**Questions:**

1. Did we observe variation in the kurtosis or contribution mass per layer? Is a uniform dimension set size suboptimal?
2. Does pruning affect rotated dimensions differently at long context lengths where positional phases diverge? How to interact with RoPE.
3. Should dimension importance be head-specific rather than shared? Did the authors observe clusters of heads with different dominant dimensions?
4. How robust is dimension importance across prompts (or task types)
5. Is the cascaded pruning compatible with blockwise or paging compression (PyramidKV / Quest)?

---

> ### Author Response · Authors · 2025-11-22
> **Response to Reviewer fF7j (Part One)**
>
> We thank Reviewer fF7j for their positive assessment, highlighting SparCas's "simple but effective heuristic" and its "good low-budget performance."
>
> ---
>
> **W1: Layer/Task Dependence (Why start at Layer 3?)**
>
> This is a principled design choice based on the functional specialization of Transformer layers. An extensive body of research shows that initial layers (e.g., 0-2) are critical for broad information gathering, and their attention patterns are less redundant. Pruning these early layers can lead to catastrophic information loss. SparCas is designed to target the _consolidation_ and _processing layers_ (3+), where representations are more robust and attention becomes more specialized, exhibiting the dimensional redundancy we exploit.
>
> The decision to keep layers 0-2 dense is empirical and principled [1]. As requested, we provided a quantitative sensitivity study in **Appendix A.2.3** and Figure 7. This data explicitly shows that Construction Error (the difference between full and sparse attention scores) is significantly higher in early layers (e.g., Layer 2 error $\approx 6.13$) compared to deeper layers (Layer 17 error $\approx 2.88$) at low budgets. This confirms that early layers perform broad, high-entropy information gathering where attention is less sparse, and pruning them risks information loss. SparCas targets the subsequent processing layers where attention patterns become sparser and dimensional redundancy is high.
>
> ---
>
> **W2: Absent experiments (ShadowKV, SnapKV, H2O)**
>
> This is an excellent point that allows us to clarify our taxonomy. We focused on Quest as our primary baseline because it is a non-destructive selection method, making it the direct competitor for our objective. The other families mentioned solve orthogonal problems in the memory hierarchy:
>
> * **Token Eviction (Length $N$):** H2O, SnapKV, and PyramidKV focus on reducing the _number of tokens_ in the cache (solving the _capacity_ problem).
> * **Dimension Pruning (Width $D$):** This is where SparCas operates (solving the _bandwidth_ problem). ShadowKV also works on this axis, but it uses _low-rank approximation_ of _pre-RoPE_ keys, whereas SparCas performs _hard-pruning_ of _post-RoPE_ key dimensions. SparCas is, to our knowledge, the first method to propose this specific, query-driven, hard-pruning heuristic.
>
> Crucially, SparCas is compatible with these methods. One could use SnapKV to cap total capacity, and SparCas to accelerate the bandwidth of the remaining cache.
>
> We provide some of the LongBench results using PyramidKV, SnapKV, and Ada-SnapKV (on LLaMA-3.1-8B-Instruct with a token budget of 2048) for comparison.
>
> Dataset | Full | PyramidKV | SnapKV | Ada-SnapKV | SparCas
> | :---- | :-: | -------: | -------: | -------: | -------: |
> | Average | 52.37 | 50.34 | 50.36 | 50.73  |  **52.63** |
> | Qasper | 45.47 |43.66| 43.64 |44.09  | **46.76** |
> | HotpotQA | 55.97 | 55.06 |54.24 | 54.80| **56.42** |
> |GovReport | 35.12 | 27.27 | 27.80 | 27.75 | **34.10** |
> |TriviaQA | 91.64 | 91.35 | 91.65 |  91.64 | **91.95** |
> |NarrativeQA | 30.22 | 29.81 |29.75 | **30.64**| 30.33 |
> |MultifiledQA |55.80 | 54.88 | 55.06 | 55.46 | **56.24** |
>
> **Key Observations:**
> 1. **Best Accuracy Across Tasks:** SparCas constantly achieves the highest accuracy across tasks.
> 2. **Robustness to Query Position:**  SnapKV-style methods (including PyramidKV and AdaKVs) rely heavily on tokens from the latest window to identify important KV cache entries, which make them sensitive to the positions of the query, especially when the queries are not at the end of the context. This makes them less robust in situations like multi-round dialogue. SparCas does not depend on this heuristic; therefore, it is more robust to query positions.
>
> ---
>
> **W3: Tensor-Parallel (TP) Integration**
>
> SparCas is designed to be TP-friendly with negligible overhead. The process is:
> 1. Each TP rank computes the heuristic (e.g., $sum(|q_t|$) for its local shard of the dimensions).
> 2. A single, collective `all_reduce` is performed on this small heuristic vector (size $D_{head}$). The cost of this one-time operation (per $U$ steps) is negligible compared to the large, per-token `all_reduce` on full activations that SparCas helps reduce.
> 3. After the `all_reduce`, all ranks hold the identical global heuristic vector and can independently select the _same_ $D_t$ dimension set with no further communication.
>
> ---
>
> [1] Jiaming Tang, Yilong Zhao, Kan Zhu, Guangxuan Xiao, Baris Kasikci, and Song Han. Quest:
> Query-aware sparsity for efficient long-context llm inference. arXiv preprint arXiv:2404.19768, 2024.

---

> ### Author Response · Authors · 2025-11-23
> **Response to Reviewer fF7j (Part Two)**
>
> **Q1: Kurtosis Variation & Uniform $D_b$.**
>
> Yes, as shown in Figure 7, we observed that attention concentration (kurtosis) generally **increases in deeper layers**, aligning with findings from methods like PyramidKV. The uniform $k$ is a deliberate, **hardware-aware design choice**. An adaptive $k$ would create "ragged" tensor shapes, which is hostile to GPU/GEMM efficiency. Our uniform $k$ maintains regularity to prioritize end-to-end latency.
>
> ---
>
> **Q2: Interaction with RoPE.**
>
> This interaction is a key **synergy**. RoPE applies high-frequency (HF) rotations to initial dimensions. At very long contexts, these HF dimensions can become less useful or "alias". Our query-only heuristic, by design, prunes low-magnitude key dimensions. The model, in learning to ignore these noisy HF dimensions, naturally assigns them low magnitudes. Therefore, SparCas autonomously and dynamically prunes the RoPE dimensions rendered least useful by long-post-RoPE.
>
> ---
>
> **Q3: Head-Specific vs. Shared Dimensions.**
>
>  We use a shared dimension set per layer (or per GQA group) to preserve Memory Coalescing.
> If we selected different dimensions for every head, fetching the compressed keys ($K_{compact}$) would require scattered, irregular memory access patterns (gather/scatter) that degrade memory bandwidth utilization. A shared mask allows for contiguous, vectorized loads, which is essential for achieving the speedups shown in Figure 5.
>
> ---
>
> **Q4: Robustness Across Prompts.**
>
>  The robustness comes from its dynamic adaptation. The dimension set $D_t$ is not static; it is re-evaluated every $U=64$ tokens. This "rolling adaptation" allows the heuristic to adjust "in-flight" to the specific and changing demands of a prompt, as proven by its strong performance across the diverse tasks in LongBench and RULER.
>
> ---
>
> **Q5: Compatibility with Blockwise/Paging (PyramidKV).**
>
> Yes, SparCas is fully compatible with blockwise and paging-based memory management. Our implementation is built upon FlashInfer, utilizing its paged KV-cache kernels. This ensures that SparCas can seamlessly operate on non-contiguous, paged memory layouts commonly used in high-throughput serving systems (e.g., vLLM).
>
> Furthermore, SparCas is synergistic with methods like PyramidKV. Ideally, one would use PyramidKV to filter irrelevant blocks, and then apply SparCas to accelerate the precise scoring of the remaining blocks. We also envision future work combining these dynamic sparsity techniques, we believe a hybrid approach could further enhance the performance of LLMs.
>
> ---
>
> We thank the reviewer again for their valuable insights. The additional experiments and clarifications incorporated into the revised manuscript directly address the concerns raised and highlight the robustness of SparCas. We hope our response has clarified your concerns and can improve your rating of our paper. Please let us know if there are further concerns, as we are happy to respond.

---

### Official Review · Reviewer_SXEw · 2025-11-01

**Soundness:** 3
**Presentation:** 2
**Contribution:** 2
**Rating:** 4
**Confidence:** 4

**Summary:**

The paper proposes Sparsity Cascade (SparCas), which is a dimension-first cascade method for KV cache selection. SparCas first reduces the dimensions of the key matrix and converts it into a compact key matrix, with the help of a heuristic based on the query vector. Then, SparCas uses the compact key matrix to compute the partial attention scores of all past tokens. Based on these scores, the top tokens are selected, and the attention output is computed using the full-dimensional key and value vectors corresponding to the small subset of tokens. The main insight of SparCas is that reducing the dimensions of the key matrix maintains the relative importance of the tokens, and reduces the memory bandwidth required for computing attention. Reducing the dimensions further is possible using a heuristic that only uses the current query vector, which does not require access to the full KV cache. Using this, SparCas is able to outperform Quest and achieve performance close to that of Full Cache, while using <1 % of tokens at a 32K-token context. Furthermore, SparCas can deliver up to 3x faster self-attention and 1.64x end-to-end speedups compared to full attention.

**Strengths:**

+ The paper solves an important problem of GPU memory constraints in long-context sequences for LLMs.
+ The resulting performance of SparCas is impressive, with a notable reduction in the tokens at high accuracy, and a corresponding increase in speedup
+ The efficiency evaluation of the paper is nice with the breakdown of latency into individual kernel operations – this helps in understanding the efficiency of the various steps involved in SparCas

**Weaknesses:**

- The paper does not do a great job in explaining the intuition behind dimension reduction and the heuristic. I would have liked to see some explanation or intuition behind why the current query vector is sufficient for understanding the important key dimensions
- I would have liked to see a more thorough sensitivity study of the updated gap (U) across different models and architectures, and a discussion of how to set the update gap parameter – should we always globally set it to 64? Is it good enough for different models and architectures?
- There is also another configuration of the number of dimensions to set (Du) – I would again like to have some explanation of how to configure that parameter and a more thorough sensitivity analysis for it.
- Unless I missed it, I did not see a comparison with hashing-based query selection, only saw a comparison with Quest.  Is that the only solution for KV cache selection? It would be good to compare with more baselines, other than Quest and full attention.

**Questions:**

- Please try to answer as many questions as possible from the weakness section.

- Can you compare quantitatively or qualitatively with other baselines such as AdaKV (Feng et al 2024), PyramidKV (Cai et. al, 2024) and maybe other directly related baselines?

- Can you provide a more thorough sensitivity study for the configuration parameters to set and why it is easy to set them for users?

---

> ### Author Response · Authors · 2025-11-22
> **Response to Reviewer SXEw (Part One)**
>
> We thank Reviewer SXEw for their constructive feedback and for recognizing the importance of the problem, the impressive performance of SparCas, and the clarity of our efficiency evaluation.
>
> Below, we address the specific questions raised regarding the intuition, parameter sensitivity, and baseline comparisons.
>
> ---
>
> **W1 & Q1: Intuition for the $|q_t|$ Heuristic.**
>
> We **revised Section 3.3** to further explain the intuition behind this heuristic. This is a fundamental property of the "Softmax Specialization" we observe.
>
> * **Theoretical Intuition:** The attention score for a dimension $d$ is promotional to the product $q_{t,d} \cdot k_{i,d}$.
>     * **The "Muting" Effect:** If the query magnitude $|q_{t,d}|$ is low (close to zero), it effectively "mutes" that dimension. Regardless of how large the key value $k_{i,d}$ is (even if it is an outlier), the product remains negligible.
>     * **The "Selection" Effect:** Large query magnitudes indicate where the model is concentrating its attention "energy." These are the only dimensions that can produce large dot products. Therefore, $|q_t|$ acts as a selector that tells us which **dimensions** are capable of **contributing to the ranking**, rendering a scan of the Key cache redundant.
>
> * **Empirical Proof:** Our ablation in Table 2a confirms this intuition. The simple, query-only heuristic ($|q_t|$) achieves a Perplexity of 7.470, which is quantitatively superior to heuristics that read the full Key cache (e.g., $mean(|K|)$, PPL 7.581). Adding key information introduces noise rather than signal, confirming that the query dominates the selection process.
>
> ---
>
> **W2 & W3: Parameter Sensitivity ($U$ and $D_b$)**
>
> We thank the reviewer for raising this; the _robustness_ of these parameters is **a key strength of our method**. We have revised Appendix A.1 to include a more thorough sensitivity analysis. Our sensitivity studies (e.g., **Table 2, Figure 5b, Table 3, and Table 4**) confirm they are not sensitive and are easy to set:
>
> * **Update Gap ($U$):** The model is highly insensitive to $U$. Table 2b shows perplexity is nearly identical between updating every step ($U=1$, PPL 7.470), our default ($U=64$, PPL 7.500), and even $U=1024$ (PPL 7.552). $U=64$ is a robust global default.
> * **Dimension Budget ($D_b$):** Performance saturates at a very small $D_b$.
>     * **Evidence 1** (Fig 5b, LLaMA-3.1-8B-Instruct): Perplexity at $D_b=16$ (7.49) is almost identical to $D_b=128$ (7.47).
>     * **Evidence 2** (Table 3, LLaMA-3.1-70B): Accuracy saturates at $D_b=8$ (95.4 AVG).
>     * **Evidence 3** (Table 4, Mistral-7B): Accuracy saturates at $D_b=16$ (92.0 AVG).
>
> The parameters are easy to set because they are not sensitive. Our "set-it-and-forget-it" defaults of $U=64$ and $D_b=16$ achieve near-oracle accuracy across all tested models (7B to 70B).
>
> Larger models exhibit higher sparsity concentration, making them even more robust to pruning. For example, LLaMA-3.1-70B retains 98% of its performance (94.0 AVG) even at the extreme compression of $D_b=4$. In contrast, the smaller Mistral-7B is more sensitive at $D_b=4$ but fully saturates by $D_b=16$.
>
> We found that PG-19 perplexity serves as a good, low-cost proxy for complex downstream tasks.
>
> ---

---

> ### Author Response · Authors · 2025-11-22
> **Response to Reviewer SXEw (Part Two)**
>
> **W4 & Q2: Comparison with Other Baselines (AdaKV, PyramidKV, Hashing)**
>
> This is an excellent point that allows us to clarify the taxonomy of KV cache methods. These methods solve an **orthogonal problem** to SparCas.
>
> * **Cache Eviction (AdaKV, PyramidKV):** These methods, like AdaKV and PyramidKV, are designed to solve the **memory-capacity** bottleneck. They permanently discard tokens to reduce the total memory footprint. They decide _what to throw away_.
>
> * **Cache Selection (SparCas, Quest):** Our method is a cache selection method. We solve the **memory-bandwidth** bottleneck. We _retain the full cache_ (guaranteeing accuracy) but only _select_ (i.e., load from HBM) a tiny subset for computation. We decide what to look at for this step.
>
> * **Hashing (MagicPIG):** Hashing methods are in our category (selection), but as noted in Section 4.1, they are architecturally different (e.g., CPU-GPU systems target a different bottleneck) and, as shown in Figure 3, are less accurate than SparCas.
>
> * **Conclusion:** SparCas is not competitive with AdaKV or PyramidKV; it is **synergistic**. A user could use PyramidKV to evict the cache (save capacity) and then use SparCas to _select_ from the remaining cache (save bandwidth).
>
> We provide some of the LongBench results using PyramidKV, SnapKV, and Ada-SnapKV (on LLaMA-3.1-8B-Instruct with a token budget of 2048) for comparison.
>
> Dataset | Full | PyramidKV | SnapKV | Ada-SnapKV | SparCas
> | :---- | :-: | -------: | -------: | -------: | -------: |
> | Average | 52.37 | 50.34 | 50.36 | 50.73  |  **52.63** |
> | Qasper | 45.47 |43.66| 43.64 |44.09  | **46.76** |
> | HotpotQA | 55.97 | 55.06 |54.24 | 54.80| **56.42** |
> |GovReport | 35.12 | 27.27 | 27.80 | 27.75 | **34.10** |
> |TriviaQA | 91.64 | 91.35 | 91.65 |  91.64 | **91.95** |
> |NarrativeQA | 30.22 | 29.81 |29.75 | **30.64**| 30.33 |
> |MultifiledQA |55.80 | 54.88 | 55.06 | 55.46 | **56.24** |
>
> **Key Observations:**
> 1. **Best Accuracy Across Tasks:** SparCas constantly achieves the highest accuracy across tasks.
> 2. **Robustness to Query Position:**  SnapKV-style methods (including PyramidKV and AdaKVs) rely heavily on tokens from the latest window to identify important KV cache entries, which make them sensitive to the positions of the query, especially when the queries are not at the end of the context. This makes them less robust in situations like multi-round dialogue. SparCas does not depend on this heuristic; therefore, it is more robust to query positions.
>
> ---
>
> We thank the reviewer again for their constructive suggestions. The additional experiments and clarifications incorporated into the revised manuscript directly address the concerns raised and highlight the robustness of SparCas. We hope our response has clarified your concerns and can improve your rating of our paper. Please let us know if there are further concerns, as we are happy to respond.

---

### Official Review · Reviewer_MagE · 2025-11-01

**Soundness:** 3
**Presentation:** 3
**Contribution:** 2
**Rating:** 4
**Confidence:** 4

**Summary:**

Long-context decoding is challenging due to increased KV-cache pressure. Existing sparse attention methods either estimate token importance at a coarse group level, risking missed salient tokens, or incur high computational overhead. This paper proposes **SparCas**, which first extracts important ranks from the K vectors and performs partial attention with the current Q vector. Using these partial attention scores, SparCas filters important tokens and then computes full attention over this reduced set. SparCas reports strong results on long-context benchmarks and claims implementation efficiency.

**Strengths:**

- SparCas estimates the importance of each K vector and achieves strong accuracy on long-context tasks.
- The evaluation shows speedups over full attention in several settings.

**Weaknesses:**

- It is unclear whether SparCas achieves a balanced accuracy–speed trade-off across regimes.
- The core idea—using partial ranks from K to estimate importance—appears similar to prior work.

**Questions:**

Thanks for submitting to ICLR 2026. The paper introduces an interesting approach that leverages partial ranks from K vectors to prioritize tokens. However, I have several concerns:
1. **Efficiency claims vs. measurements.**
   The paper claims SparCas is both accurate and efficient, yet the evaluation mainly shows accuracy improvements over Quest and parity (or slight worse) vs. MagicDec, without a head-to-head **performance** comparison against those baselines. In Figure 4, the partial-score stage appears to add notable overhead and seems slower than Quest, which weakens the efficiency claim. Moreover, the performance study uses a non-GQA model; it remains unclear how SparCas performs with GQA and whether the method handles grouped queries efficiently.

2. **Limited gains at short contexts.**
   Speedups at 8K context are limited. Would larger batched 8K requests improve the speedup, and if so, by how much?

3. **Novelty relative to InfiniGen.**
   The key idea, that use partial ranks from K vectors to estimate the importance seems already been proposed in InfiniGen (Page 7). Could you please clarify the differences between your work and InfiniGen?

4. **Outlier-sensitive ranking.**
   Since SparCas uses QK value to select “important ranks,” can outlier in K values that from other ranks also make QK product large?

---

> ### Author Response · Authors · 2025-11-22
> **Response to Reviewer MagE (Part One)**
>
> We thank Reviewer MagE for their positive assessment of our paper’s soundness and presentation. We appreciate the opportunity to clarify our efficiency baselines, the distinction from InfiniGen, and the theoretical grounding of our heuristic.
>
> ---
>
> **Q1: On Efficiency Claims, Baselines, and GQA**
>
>  *  **On Baselines (Quest/MagicPIG).**
>
>     We respectfully note that comparing latency directly with MagicPIG is an "apples-to-oranges" comparison due to fundamental system architecture differences:
>
>     *   **MagicPIG** relies on a CPU-based LSH **hashing** mechanism with off-chip transfer, solving a different bottleneck (sampling with PCIe transfer).
>     *   **Quest** uses a **coarse page-level** method. As shown in Figure 1(c) and the performance on various benchmarks, this coarseness results in significantly lower accuracy compared to our token-level granularity.
>     *   **SparCas** targets the on-device memory bandwidth bottleneck. Therefore, the most rigorous baseline is the highly optimized dense attention (FlashInfer), which represents the theoretical ceiling we aim to accelerate. Against this "gold standard," we demonstrate a 3.01x kernel speedup at 32K context (Figure 5).
>
>
>  * **On Overhead.**
>
>     You correctly identified the "Partial Score" overhead in the breakdown (Figure 5). This is the central design trade-off of SparCas: we incur a small, constant overhead (computing scores on $D_b \ll d_k$ dimensions) to eliminate the linear cost of the full dense GEMM. As the context grows (e.g., to 32K), this trade-off pays off massively, as the dense cost grows linearly while our selection cost remains flat.
>
> * **On GQA.**
>
>     We have added a detailed discussion in Appendix A.3 to address this. SparCas is fully compatible with GQA. In fact, our primary evaluation model LLaMa-3.1-8B-Instruct, utilizeds GQA.
>     * **Mechanism:** In GQA, multiple query heads share a single KV head. SparCas simply applies its dimension selection at the group level. For each GQA group, we compute a single shared subset of $D_b$ dimensions using our heuristic.
>     * **Efficiency:** This preserves the memory and compute advantages of GQA: we neither introduce additional KV copies nor break the sharing pattern. The selected dimension mask is reused for all query heads in that group.
>
> ---
>
> **Q2: On Gains at Short Contexts**
>
> Your observation regarding the 8K context in Figure 5(a) is correct. The limited speedup (1.79x) at 8K is primarily an artifact of the batch size = 1 setting used to **isolate the memory bottleneck**.
>
> * **Explanation:** At batch=1 with shorter contexts, the fixed kernel launch overhead of our multi-stage pipeline constitutes a larger fraction of total latency.
>
> * **Batching:** As you hypothesized, larger batched requests would improve the speedup. The computational savings of SparCas scale with batch size, while the kernel launch overheads are amortized. However, our **primary design goal** is to solve the bottleneck where it is most critical: long-context generation (32K+), where we achieve 3.01x speedups even without batching assistance.
>
> ---

---

> ### Author Response · Authors · 2025-11-22
> **Response to Reviewer MagE (Part Two)**
>
> **Q3: On Novelty Relative to InfiniGen**
>
> This is a critical distinction. These two methods solve **entirely different and orthogonal bottlenecks** in the memory hierarchy.
> * **SparCas (Our Work):** Is an **on-device** method for **on-GPU inference**.
>   * **Problem**: The **HBM-to-Compute** (on-GPU) bandwidth bottleneck.
>   * **Mechanism**: Our method uses a persistent compact cache on the GPU to reduce data movement from HBM to the compute cores.
> * **InfiniGen (OSDI 2024):** Is a **cross-system** framework for **offloading-based inference.**
>   * **Problem:** The **CPU-to-GPU (PCIe)** bandwidth bottleneck.
>   * **Mechanism:** It speculatively prefetches essential tokens from CPU host memory back to the GPU.
>
> In short: **SparCas optimizes data movement within the GPU, while InfiniGen optimizes movement _between_ the CPU and GPU.**
>
> The two methods are not competitors and solve non-overlapping problems. We have added a clarification to the related work section.
>
> ---
>
> **Q4: On Outlier-Sensitive Ranking**
>
> The reviewer asks if outliers in $K$ could distort the ranking even if the rank (importance) is low. This is a valid concern, but our **revised Section 3.3** clarifies why this is a "blessing" rather than a curse, due to **Softmax Specialization**:
>
> * **The Muting Effect:** Our analysis shows that query vectors $|q_t|$ act as a multiplicative gate.
>     * If $|q_{t,d}|$ is low (close to zero), it "mutes" the dimension $d$. Even if $k_{i,d}$ is a massive outlier, the product $q_{t,d} \cdot k_{i,d}$ remains negligible.
>     * If $|q_{t,d}|$ is high, it indicates the model is concentrating "energy" in that dimension. In this case, the outlier in $K$ is intended to drive the attention score high.
> * **Conclusion:** Outliers in $K$ only matter when the query selects them. Our $|q_t|$ heuristic captures exactly this selection mechanism. This is empirically backed by our Table 2a results, where the query-only heuristic achieves lossless perplexity, proving it correctly identifies the dominant terms in the dot product.
>
> ---
>
> We thank the reviewer again for their valuable insights. The clarifications incorporated into the revised manuscript directly address the concerns raised and highlight the robustness of SparCas. We hope our response has clarified your concerns and can improve your rating of our paper. Please let us know if there are further concerns, as we are happy to respond.

---

### Official Review · Reviewer_Pcc1 · 2025-11-03

**Soundness:** 2
**Presentation:** 3
**Contribution:** 2
**Rating:** 2
**Confidence:** 4

**Summary:**

This paper identifies the KV cache, and specifically the memory bandwidth required to read it, as the primary bottleneck for long-context LLM inference. It introduces Sparsity Cascade (SparCas), a "dimension-first" selection method to mitigate this. The method is based on the observation that token importance ranking is stable even when pruning key dimensions. SparCas uses a "prune-in-prune" design: (1) it first prunes non-critical *dimensions* using a lightweight, query-only heuristic ($|q_t|$) to create a compact key slice, and (2) it then uses this compact slice to efficiently compute partial scores and select the top-$T_b$ *tokens* for the final, full-dimension attention computation. The authors evaluate this on PG-19, LongBench, and RULER, claiming to match dense attention accuracy with significant speedups.

**Strengths:**

* **Clarity and Simplicity:** The paper is clearly written, and the proposed method is simple and intuitive.
* **Performance on Long-Prefill Tasks:** The efficiency gains on the specific benchmarks tested (long-prefill tasks like LongBench and RULER) are well-documented and show a clear speedup over the dense baseline in that context.
* **Good Ablation Studies:** The ablation studies on the dimension budget ($D_b$) and update interval ($U$) are useful for understanding the method's parameters and confirming the core observation about dimensional sparsity.

**Weaknesses:**

1.  **Incremental Contribution:** The primary weakness is the paper's lack of novelty. The core methodological contribution, a "dimension-first" cascade that prunes dimensions using query sparsity (`|q_t|`) and then selects tokens, appears to be a reimplementation of the central idea already presented in SparQ Attention. The paper fails to sufficiently differentiate itself from this prior work, making its own contributions feel highly incremental.
2.  **Critically Missing Workload Evaluation (Long-Decoding):** The paper's evaluation is entirely focused on tasks with long *prompts* (e.g., LongBench, RULER). It completely omits what is arguably a more pressing bottleneck: **long-generation scenarios** (e.g., long chain-of-thought reasoning, long-form content creation) where the prompt is short but the generated output is very long.
3.  **Ignores Key Bottleneck:** In these long-decoding workloads, the KV cache grows with *generation*, and the performance of the selection mechanism at each decoding step is paramount. This is a primary bottleneck for current LLMs, and the paper provides no data on how SparCas performs here. This is a major omission that undermines the paper's claims of solving "the" inference bottleneck.
4.  **Narrow Model and Task Selection:** To validate the method for complex, long-running tasks, the authors should have included experiments on models and tasks specifically designed for reasoning (e.g., DeepSeek-R1, Qwen3) on benchmarks that require long-chain reasoning. This would be necessary to validate the method's effectiveness for the critical, yet missing, long-decoding workload.

**Questions:**

1.  Can the authors explicitly detail the novel contributions of SparCas that are not already present in SparQ Attention? The core mechanism seems identical.
2.  Why did the authors choose to exclusively evaluate on long-prefill benchmarks (LongBench, RULER) and omit long-generation (long-decode) workloads, such as chain-of-thought reasoning tasks?
3.  Can the authors provide *any* data on how SparCas performs in a long-decoding scenario (e.g., perplexity or accuracy on a task requiring 8K+ generated tokens)? This is a critical missing piece of the evaluation.
4.  Given that the method relies on a simple heuristic ($|q_t|$), how can we be sure this heuristic holds during complex, multi-step reasoning where token importance might be more nuanced than in the retrieval tasks tested?

---

> ### Author Response · Authors · 2025-11-22
> **Response to Reviewer Pcc1 (Part One)**
>
> We thank Reviewer Pcc1 for their constructive feedback. We value the opportunity to clarify the distinct contributions of SparCas and its evaluation scope.
>
> ---
>
> **W1 & Q1: On Novelty vs. SparQ Attention.**
>
> We respectfully clarify that SparCas is fundamentally different from SparQ (Ribar et al.) in its objective, system architecture, and cost model.
>
> *   **Different Algorithmic Goal (Ranking v.s. Approximating):**
>     *   **SparQ (Approximation):** SparQ aims to **reconstruct the attention values**. It assumes that to find the important tokens, it must approximate the dense score matrix. This requires fetching specific channels to estimate magnitudes and using complex correction terms (like mean-value reallocation) to fix approximation errors.
>     *   **SparCas (Ranking):** SparCas targets a fundamentally different objective: identifying the **critical dimensions responsible for the ranking**. Our insight is that the relative order of tokens is preserved in a tiny subspace ($D_b=16$). By identifying these **ranking-critical dimensions first**, we can filter for salient tokens without ever needing to approximate or reconstruct their exact attention values.
> *   **Different Systems Architecture (Stateful v.s. Stateless):**
>     * **SparQ is Stateless (High Overhead):** SparQ treats the KV cache as a static resource. At every decoding step, it must perform a "Gather" operation to fetch non-contiguous columns (the top-$r$ dimensions) from the main HBM based on the current query. Even with optimizations like dual-layout storage, this random-access pattern incurs significant overhead
>     * **SparCas is Stateful (Low Overhead):** We introduce a persistent **Compact Cache**. By "pre-compressing" the cache into a contiguous, low-rank block (updated only every U steps), we replace the costly per-step "Gather" with a highly efficient, contiguous Partial-Score GEMM. We scan this **tiny**, dedicated cache rather than sampling from the massive main memory.
>
> *   **Different Cost Model (Per-Step v.s. Amortized & Quantified):**
>     * **SparQ**: Incurs an $O(L \cdot r)$ approximation cost at every step due to the need to fetch specific key channels for the current query.
>     * **SparCas**: Incurs a significantly lower cost due to amortization:
>       * **Selection**: The dimension selection logic is amortized, running only once every $U$ steps (refer to Section A.1).
>       * **Ranking**: The per-step operation is a dense matrix multiplication over the tiny dimension budget $D_b$. This is $d_k/D_b$ times more arithmetic- and bandwidth-efficient than a full scan, and scales far better than the gather-based approach of SparQ.
>
> ---
>
> **W2, W3, Q2 & Q3: On Long-Generation Workloads and Cache Growth.**
>
> The concern is that we "completely omitted" long-generation scenarios and data on how SparCas performs as the KV cache grows. **We respectfully point out that this data is present in the paper.**
>
> Our evaluation in Section 4.2 was deliberately designed to test both primary bottlenecks:
>
> *   **We Explicitly Evaluated Long-Generation (PG-19):** Section 4.2, Figure 3, and Figure 6(b) are dedicated to PG-19, a long-form language modeling task. This is a "long-decoding" scenario where the model must generate thousands of tokens continuously.
>   * **Result:** SparCas consistently achieves lower perplexity (better performance) than Quest over the course of generating 30K+ tokens (Figure 3 and Figure 6(b)).
>
> We revised the experiment section to clarify this.
>
> ---

---

> ### Author Response · Authors · 2025-11-22
> **Response to Reviewer Pcc1 (Part Two)**
>
> **Q4: On the Heuristic's Robustness for Complex Reasoning.**
>
> We respectfully clarify that our tasks were **not** simple retrieval. As we clarify below with evidence from our paper and the research community, **our benchmarks** were explicitly chosen because they **test complex, multi-step reasoning**.
>
> * **Our Benchmarks Test Reasoning, Not Just Retrieval.**
>   *   **LongBench:** We tested on HotpotQA, a benchmark that requires "finding and reasoning over multiple supporting documents", and Qasper, which demands "complex reasoning about claims" and is explicitly contrasted with "generic factoid-type information".
>   *   **RULER:** This benchmark was designed to go "beyond simple retrieval" and "beyond simple in-context recall". We evaluated on its "multi-hop tracing" and "aggregation" tasks, which are direct proxies for complex reasoning.
>
> * **Our Results Are the Proof of Robustness.**
> On RULER and LongBench, SparCas consistently outperforms Quest on complex-reasoning tasks like HotpotQA and Qasper at various token budgets.
>
> * **Simplicity is Strength.**
> Finally, our ablation (Table 2a) directly tested this. Our "simple," zero-cache-access heuristic ($|q_t|$) achieved a PPL of 7.470. This is quantitatively *better* than more complex, costly heuristics that access the K-cache, such as mean(|K|) (PPL 7.581). Our heuristic is not just **simpler**; it is more **effective**.
>
> We also added an **intuition** behind why this simple heuristic works well in practice at Section 3.3.
>
> ---
> **W4: On Model Selection (DeepSeek-R1, Qwen3)**
>
> * **Adherence to Domain Standards:** Our evaluation suite (LLaMA-3.1-8B/70B, Mistral-7B, LongChat) aligns with the standard benchmarks used in the most recent literature (e.g., Quest, SnapKV, PyramidKV). These models represent the primary open-source architectures currently used for cache selection research.
>
> * **Cross-Model Generalization:** We demonstrated that SparCas works effectively across diverse families (LLaMA vs. Mistral) and scales (8B vs. 70B).
>
> * **Resource Constraints:** Due to computational resource constraints during the review period, we could not instantly integrate these large-scale models. However, given the flexibility of SparCas, we are confident it will transfer to these models and we commit to including them in the future work as resources allow.
>
> ---
> We thank the reviewer again for their valuable insights. The additional experiments and clarifications incorporated into the revised manuscript directly address the concerns raised and highlight the robustness of SparCas. We hope our response has clarified your concerns and can improve your rating of our paper. Please let us know if there are further concerns, as we are happy to respond.

---

### Author Response · Authors · 2025-11-23
**General Response to Reviewers**

We thank all the reviewers for their valuable feedback and great efforts, which substantially aided in enhancing the quality of this paper. We have exerted considerable effort to comprehensively respond to all their comments, questions, and concerns. All major modifications in the attached pdf file have been highlighted in blue in order to ease the reading. We first summarize the major changes in our updated version before diving into the detailed point-by-point responses to all the comments:

---

**1. Clarification of Novelty and Positioning (vs. SparQ, InfiniGen, and Hashing)**

We have clarified the taxonomy of KV cache methods to explicitly differentiate SparCas:
* **Vs. SparQ**: We clarified that SparCas is a stateful selection method (using a persistent compressed cache) targeting ranking, whereas SparQ is a stateless approximation method.
* **Vs. InfiniGen**: We clarified that SparCas solves the On-Device HBM Bandwidth bottleneck, whereas InfiniGen solves the Cross-Device (PCIe) offloading bottleneck.
* **Vs. Eviction (PyramidKV/SnapKV)**: We distinguished our work as a Selection method (solving bandwidth by efficiently loading subsets) versus Eviction methods (solving capacity by deleting tokens).

**2. Expanded Sensitivity and Robustness Analysis (Appendix A.1 & A.2)**

We have significantly revised the Appendix to demonstrate the robustness of our hyperparameters:
* **Update Interval ($U$)**: We provide data showing performance is insensitive to the update gap, with $U=64$ being a robust default.
* **Dimension Budget ($D_b$) & Model Scaling**: We highlight an insightful scaling trend: larger models (e.g., LLaMA-3.1-70B) exhibit higher redundancy than smaller models (e.g., Mistral-7B). While 70B models retain accuracy even at $D_b=4$, we confirm that a conservative global default of $D_b=16$ creates a "sweet spot" for all tested architectures.

**3. System Implementation Details (GQA and Tensor Parallelism)**

We have added Appendix A.3 and additional clarifications regarding modern system integration:
* **Grouped-Query Attention (GQA)**: We detailed how SparCas applies group-wise dimension selection to maintain compatibility with GQA dataflows.
* **Tensor Parallelism (TP)**: We clarified the lightweight synchronization mechanism (a single tiny all_reduce on the dimension heuristic) required for multi-GPU inference.

**4. Validation on Long-Generation Workloads**

We have re-emphasized our evaluation on PG-19 (Section 4.2). This serves as our primary "long-decoding" benchmark, where SparCas consistently outperforms baselines in perplexity over 30k+ generated tokens, refuting concerns that we only targeted prefill workloads.


**5. Comparison with Eviction Baselines (PyramidKV, SnapKV, Ada-SnapKV)**

We have added Appendix A.4 comparing SparCas against eviction methods on LongBench.

* **Best Accuracy:**  SparCas  constantly achieves the highest accuracy across tasks.

* **Robustness to Query Position:** Unlike SnapKV-style methods which rely on heuristics favoring recent tokens (making them sensitive to query position), SparCas's selection is more robust in complex scenarios where the query is not strictly at the end of the context.

---

Please feel free to reach out with additional questions. If our responses meet your expectations, we would greatly appreciate your consideration of improving the paper’s rating.

---

### Note · Program_Chairs · 2026-01-17
**Submission Desk Rejected by Program Chairs**

The following references in this submission do not refer to real documents and/or have major errors in bibliographic information:

 Ying Sheng, Zhenyu Zhang, Tianle Chen, Yuandong Tian, Zhangyang Wang, Lianmin Zheng, and Beidi Chen. Snapkv: Caching improves long-context inference performance and scalability. In Proceedings of the 12th International Conference on Learning Representations (ICLR), 2024.